# X-ray Computed Tomography Method for Macroscopic Structural Property Evaluation of Active Twist Composite Blades

**Joon H. Ahn** [1], **Hyun J. Hwang** [1], **Sehoon Chang** [1], **Sung Nam Jung** [1,\*], **Steffen Kalow** [2] **and Ralf Keimer** [2]

1   Department of Aerospace Information Engineering, Konkuk University, 120 Neungdong-ro, Gwangjin-gu, Seoul 05029, Korea; joonhyekahn@gmail.com (J.H.A.); zammanboh@gmail.com (H.J.H.); pattern@konkuk.ac.kr (S.C.)
2   German Aerospace Center (DLR), Institute of Composite Structures and Adaptive Systems, 38108 Braunschweig, Germany; Steffen.Kalow@dlr.de (S.K.); Ralf.Keimer@dlr.de (R.K.)
\*   Correspondence: snjung@konkuk.ac.kr

**Abstract:** This paper describes an evaluation of the structural properties of the next-generation active twist blade using X-ray computed tomography (CT) combined with digital image processing. This non-destructive testing technique avoids the costly demolition of the blade structure. The CT scan covers the whole blade region, including the root, transition, and tip regions, as well as the airfoil blade regions, in which there are spanwise variations in the interior structural layout due to the existence of heavy instrumentation. The three-dimensional digital image data are processed at selected radial stations, and finite element beam cross-section analyses are conducted to evaluate the structural properties of the blade at the macroscopic level. The fidelity of the digital blade model is first assessed by correlating the estimated blade mass with the measured data. A separate mechanical measurement is then carried out to determine the representative elastic properties of the blade and to verify the predicted results. The agreement is found to be good to excellent for the mass, elastic axis, flap bending, and torsional rigidity. The discrepancies are less than 2.0% for the mass and elastic axis locations, and about 8.1% for the blade stiffness properties, as compared with the measured data. Finally, a sensitivity analysis is conducted to clarify the impact of modeling the sensor and actuator cables, nose weight, and manufacturing imperfections on the structural properties of the blade.

**Keywords:** active twist blade; CT scan; digital reproduction; structural property; sensitivity analysis

## 1. Introduction

Conventional helicopters are prone to high levels of vibration due to the edgewise condition of the rotor in flight. A variety of engineering approaches and active rotor technologies have been devised to alleviate this vibration. One promising concept is the active twist rotor (ATR) scheme, in which the activation of piezoceramic fiber composites built into the blade structure enables the individual pitch angles to be varied according to the desired control effect [1–4]. The main advantage of ATR technology over other actuation schemes is that the actuator itself is used as a primary load-bearing structure without resorting to any amplification devices. Chen and Chopra [5] examined the strain-induced actuation concept for the generation of the twist deformation via thin monolithic piezoceramic actuators embedded in the composite skin layers. Despite the innovative concept, the resulting actuation strokes remained so small (tip twist amplitudes of about 0.1° in the hover condition) that this ATR scheme was far from being applicable in real-world scenarios. The advent of active fiber composite (AFC) and macro-fiber composite (MFC) material technologies enabled piezo materials to be fabricated into fibers [6,7] and rectangular sheet forms [8] respectively, and there have since been several attempts to explore the benefits of the ATR concept in scaled model rotor applications [7,9]. Among

others, an international consortium project called STAR (Smart Twisting Active Rotor), with a team of the U.S. Army, NASA Langley, German DLR, French ONERA, Dutch DNW, the Japan Aerospace Exploration Agency (JAXA), Korea Aerospace Research Institute (KARI), and Konkuk University, has demonstrated the effectiveness of ATR for a rotor under wind tunnel conditions [10,11].

The STAR wind tunnel test was originally to be carried out at the DNW facility (German–Dutch Wind Tunnel) in 2013. During the hover test run, unexpected blade structural integrity problems associated with actuator failures (e.g., short circuits) were encountered, resulting in substantial delays to the test campaign. A careful inspection of the blade surfaces revealed that the failure was induced by the high strain level experienced near the blade trailing edge (TE) region as a result of the centrifugal action of the rotor in operation. Since then, significant modifications have been made to the blade structures [12,13]. Compared with the original blade designs, notable changes include: (1) carbon fiber-reinforced composites (CFRP) have replaced the glass fiber composites (GFRP) to provide increased tensile rigidity, (2) composite straps (TE spars) have been attached under the skin near the TE for local stiffening, and (3) the nose weight area has been enlarged to counter the rearward shift of the mass center (or center of gravity, CG) induced by the structural modifications. Overall, these changes result in significant increases in blade stiffness and weight compared with the first-generation STAR (called STAR I) blades.

The aim of the present work is to evaluate the structural properties of the refabricated AT blades (called STAR II blades) without destroying the blades. For this purpose, a non-destructive X-ray computed tomography (CT) scan technique combined with digital image processing and finite element (FE) beam cross-section analysis is applied. The basic principle behind the exploitation of CT in identifying the material distributions over the structure is that each voxel (volume element) image has a discrete CT number (called Hounsfield unit in the medical field) according to the physical density of the material domain [14]. This method was successfully used by Jung et al. [15] to determine the structural properties of the HART II blades [16,17]. CT images were available for limited portions of the blade in an earlier study [15] that focused only on the blade root region. The current study aims to cover the whole blade region (root, transition, airfoil, and tip) to ensure a comprehensive evaluation of the blade section properties. It is emphasized that STAR blades contain heavy instrumentation (constituting about 15% of the blade structural weight), particularly over the blade outboard region. Thus, the blade airfoil region cannot be regarded as uniform, unlike the HART II blades. Three-dimensional (3D) digital images of the constructed full-span blade are used to generate a two-dimensional (2D) FE cross-sectional analysis model, which allows the blade structural properties to be evaluated. Note that the present approach is different from conventional analytic methods in that the structural layouts and geometric configurations of the real post-manufactured blades (including manufacturing defects or imperfections) are taken into account as they appear in the digitally reconstructed images. The accuracy of the proposed method is assessed by correlating the estimated blade weight (including pressure sensors and connecting cables) and the representative structural properties (shear center and elastic stiffnesses) with the mechanically measured data. A design sensitivity analysis is also carried out to quantify the influence of the blade cross-sectional elements used in the model and manufacturing imperfections on the evaluation of the blade structural properties. The former include sensor and actuator cables as well as the nose weight materials, while the latter considers misfits of the TE spar elements.

## 2. Digital Reconstruction of the Blade

This section describes the basic features of STAR blades and provides detailed descriptions of how full 3D digital images of the manufactured blades are generated from the CT scan images. Finally, we describe the construction of 2D FE cross-section models for structural property evaluations.

*2.1. General Features of STAR Blades*

The STAR rotor is a four-bladed, Mach-scaled model of the production BO-105 helicopter [11]. The rotor has a solidity of 0.077 and a radius (*R*) of 2 m and chord length (*c*) of 0.121 m. The blades are of rectangular planform with a linear pre-twist distribution of –8° along the length. The general features of the STAR rotor are similar to those of its predecessor programs, the HART I [18] and HART II rotors [17]. Some notable improvements to the STAR rotor include: (1) an articulated hub rather than a hingeless configuration, (2) MFC actuator patches installed underneath the blade skin, and (3) more cable arrays and accessories. Specifically, as many as 217 pressure sensors and 30 actuator patches are installed in STAR blades, far more than in HART I (124 pressure sensors) and HART II (51 pressure sensors). Note that both HART rotors have a significant blade-to-blade dissimilarity in their measured blade responses, because only one of the blades was heavily instrumented [17]. In the case of STAR, an identical number of sensors (dummy or real) and cables are installed in each constituent blade to maintain similar static and dynamic features throughout, thereby overcoming the dissimilarity issue. Figure 1 shows a photo of the STAR blade. A number of sensor cables are knotted in groups and exposed to the air out of the blade root. The cable arrays are indeed connected to a terminal box within an aluminum housing that is used for data acquisition during test runs. The terminal box and housing are not considered in the present blade model. Nonetheless, these root attachment parts should be considered in the aeromechanics analysis stage, as they may produce significant aerodynamic loads (mostly drag) while in operation.

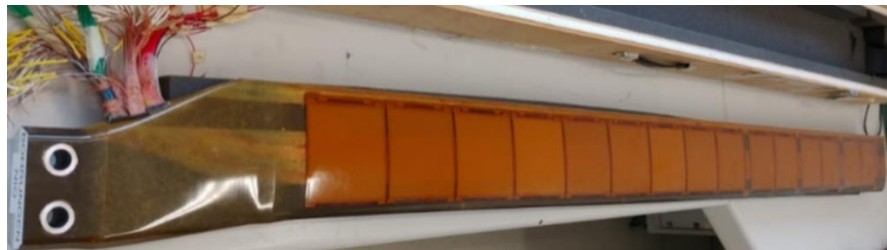

**Figure 1.** STAR blade used for X-ray CT scan and structural property evaluation.

*2.2. X-ray CT Scan and Image Processing*

The CT test facility (Phoenix V | tome | x L450 by Rober) at the Institute of Structures and Design, DLR Stuttgart, was used to obtain high-quality, full 3D digital images of the STAR blade. Figure 2 shows the CT scan setup with the STAR blade mounted on top of the spindle axis. The X-ray CT technique uses computer-processed X-rays to create tomographic images of an object. The CT scan system consists of the X-ray source (450 kV/1500 Watt), digital detector, and image processing software. First, a number of 2D projection images are constructed at different projection angles while rotating the object through the desired orientation angles. At each projection, the X-ray tube produces a narrow, cone beam of X-rays that passes through a section of the object. The detector registers the X-rays as a snapshot in the process of creating images. About 2400 X-ray projections are performed for one complete revolution of the object. Note that each projection is taken four times and averaged to improve the quality of the CT images.

The maximum scan range of the X-ray test facility is up to 800 mm (width) by 1100 mm (height), which covers about half of the span length of the STAR blade. Due to the high-resolution image quality requirement, however, the range of the scan per execution should be reduced and adjusted appropriately. Considering the slender nature of the blade, the scan range per projection was set to be ~155 mm wide, which can fit the chord (121 mm) of the blade. Then, the resulting pixel size set for the scan is 361 μm. Slightly larger dimensions than the actual chord are necessary because the blade root around the blade bolt protrudes slightly toward the leading edge (LE) (see Figure 1).

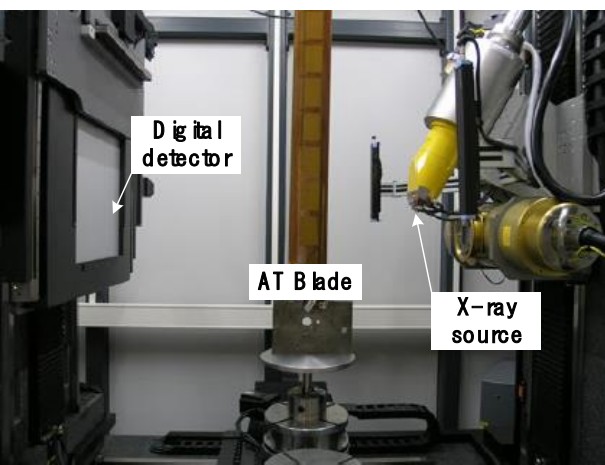

**Figure 2.** AT blade setup installed for X-ray CT scan (blade placed in the cardboard container).

Given the limitations in the hardware setup, the blade was divided into six sub-regions (S1–S6) for the CT scan, as depicted in Figure 3a. The scan covers most of the blade root and tip regions (S1–S4). In the blade airfoil region, only zones S5 and S6, which have a full array of pressure sensors along the chord, are considered. This is because there are significant deviations in the sensor cable counts at each pressure measurement location, and this affects the mass and inertia characteristics (without influencing much on the stiffness properties) of the blades. Note that the pressure was measured at 67.2%, 77.3%, 87.5%, and 97.5% of the rotor radius ($R$). The remaining portions of the blade airfoil regions are assumed to be uniform. When taking CT scans of S3–S6, the blade was held upside down due to the stroke limit of the X-ray test facility. Figure 3a also shows a planar view of the CT scan images for the respective subregions overlapped with the blade CAD drawing. An enlarged view of the blade root region (S1 and S2) is presented in Figure 3b. The detailed interior layout and different materials of the blade structure are captured nicely in gray-scale. The darker zones in the X-ray image indicate regions with low-attenuation coefficients (low density), while the brighter zones signify high X-ray absorption rates (greater material density). The respective zones inside the blade structure are identified in Figure 3b, taking into consideration the gray-scale level and the known topology of the blade structure.

Figure 4 shows some cross-sectional CT images at designated radial stations of the blade root and transition regions (R3, R5, R7), as well as the blade airfoil region (U3, U5, U7). The corresponding blade radial stations are indicated in Figure 3a. These radial stations were chosen because substantial changes in blade external geometries or internal layouts exist, and so these regions would provide a reference when performing 2D cross-sectional analysis. It is interesting to note that a fairly large number of cable arrays are present in both the blade root region and the blade airfoil region. This is not surprising considering the heavy instrumentation of STAR blades (including dummy sensors), as mentioned previously. The cable count varies significantly along the blade length. The highest number is found at the innermost station (R3), at which as many as 553 cables appear to be nested in the section: 392 for the pressure sensors, 129 for the strain gages, and 32 for the actuators. The sensor and actuator cables are made of copper-silver, with a polytetrafluorethylene (PTFE, Teflon®) coating. Due to the high material density of copper, they have a non-negligible effect on the CG location and the section inertia, as will be shown later. Finding a way of shifting the CG toward the LE was a major issue when redesigning the STAR blades [11]. The final solution was to shift most of the sensor cables, especially the pressure sensor cables, far into the airfoil LE region, along with an increased diameter of the nose weight (see Figure 4). The series of rectangular brackets in section U5 are the pressure sensor housings. Note that their post-manufactured positions in the blade are captured well in the X-ray images.

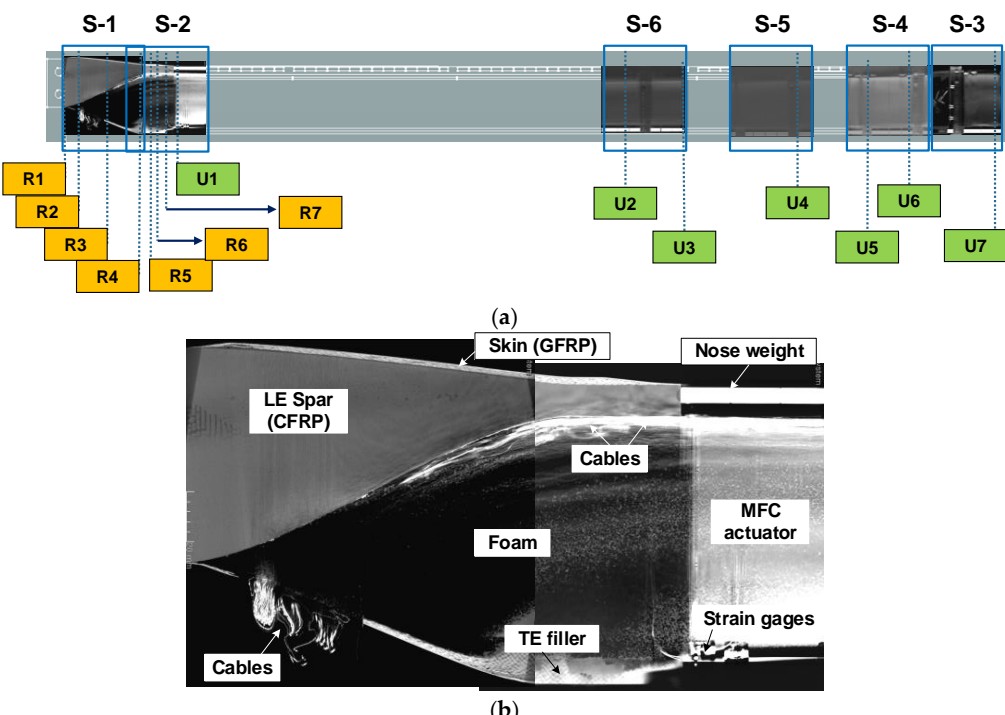

(a)

(b)

**Figure 3.** CT scan images over the blade subregions. (**a**) CT scan regions and designated blade radial stations of the STAR blade. (**b**) Enlarged view of CT scan images around the blade root regions (S1 and S2). A higher resolution image can be found in the Supplementary Materials, Figure S1.

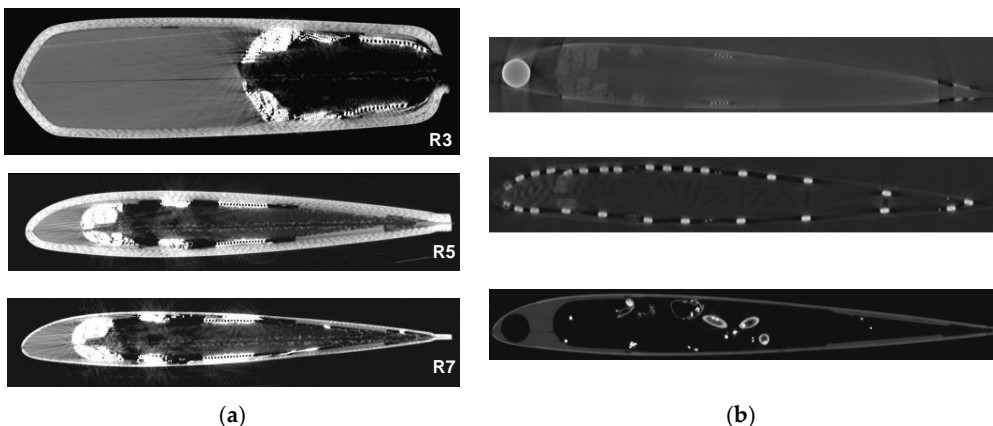

(a)                                          (b)

**Figure 4.** CT scan images at designated radial stations of the blade (not to scale). (**a**) Blade root region and (**b**) blade airfoil region. Higher resolution images can be found in the Supplementary Materials, Figures S2–S7.

Using the constructed CT images, a section segmentation process was performed to discern each subregion having identical X-ray attenuation coefficients (same material) in the blade section. The segmentation process is very labor-intensive and was performed using a standard graphics software tool (e.g., VGSTUDIO ver. 3.2.0). The resulting segmented images of two representative sections stationed at 0.154$R$ (R2) and 0.655$R$ (U2) are presented in Figures 5 and 6. The color codes in the RGB scale are used to classify each material domain of the section. For instance, an RGB color code of (0 125 255) represents the skin made of ±45° unidirectional (UD) glass-epoxy composite plies, while an RGB code of (255 255 0) indicates the foam material. All the individual sensor cables captured in the images are marked in different RGB scales. The mechanical material properties corresponding to the specific zones of the section are summarized in Table 1. For the sensor and actuator cables, a mixture rule (between copper and Teflon® skin) is used to represent

the mechanical properties. It should be emphasized that the manufacturing defects or imperfection zones (except artifacts) found in the CT images are reflected in the section segmentation process in the same way as they appear in the images. Specifically, they are depicted in dotted circles in Figures 5 and 6 as: (1) varying thickness pattern in the resin layer under the skin (Figure 5), (2) additional resin materials found at the corners in the foam material domain (Figure 5), and (3) possible misfits in TE spars (Figure 6). It is noted that the dark horizontal line and the brighter lines captured in Figure 5a are neglected in the segmentation stage, considering the feature that their impact on the structural properties should be marginal due to a small thickness in the resin layer and/or similarity in elastic properties between GFRP and CFRP composites. The incorporation of the thin layers in the model may contribute to a drastic increase in the number of degrees of freedom (number of FE's) with no significant changes. Once specific material zones of the section had been identified from the section segmentation process, each segmented region was partitioned into discrete surfaces with the aid of the commercial preprocessing software MSC.PATRAN. This led to several subregions with prescribed boundary edges each having the same material composition. The FE modeling and simulation were then conducted to discretize and analyze the corresponding portions of the blade, as described in subsequent sections.

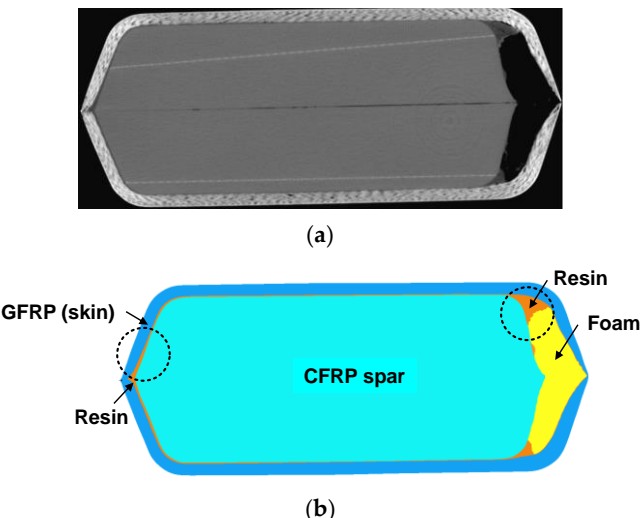

**Figure 5.** Comparison of section images at blade root station R2. (**a**) X-ray scan image and (**b**) section-segmented image.

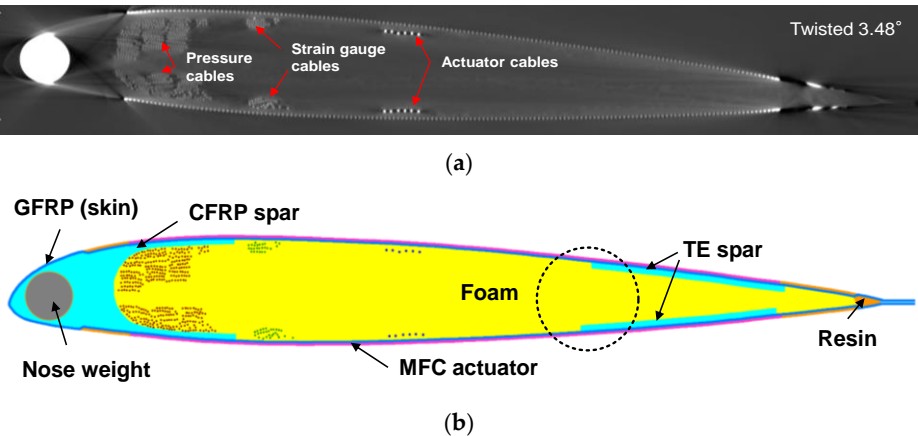

**Figure 6.** Comparison of section images at blade airfoil station U2. (**a**) X-ray scan image and (**b**) section-segmented image.

**Table 1.** Mechanical material properties.

| Material | $E_L$ [a] (GPa) | $E_T$ [b] (GPa) | $G_{LT}$ [c] (GPa) | $\rho$ [d] (kg/m³) |
|---|---|---|---|---|
| UD CFRP (spar) | 177 | 9.1 | 5.08 | 1580 |
| UD GFRP (skin) | 45.17 | 11.98 | 4.13 | 2008 |
| MFC | 30 | 15.5 | 10.7 | 4700 |
| Titanium (nose weight) | 50.2 | 50.2 | 19.6 | 15,020 |
| Epoxy resin | 3.416 | 3.416 | 1.289 | 1124 |
| Polyurethane foam | 0.075 | 0.075 | 0.025 | 52 |
| Sensor cables | 12.55 | 5.1 | 4.902 | 2168 |
| Actuator cables | 8.00 | 4.7 | 3.125 | 2048 |

[a,b,c] Elastic moduli in tension ($E$) and shear ($G$) in longitudinal (L) and transverse (T) directions. [d] Material density.

### 2.3. Two-Dimensional (2D) Cross-Sectional Analysis

A general purpose, FE-based 2D cross-sectional analysis program called KSEC2D II (Konkuk SECtion 2D [19,20]) was employed to analyze the individual cross-sections of the STAR blades. KSEC2D II is an expanded version of the earlier KSEC2D [21], and is suitable for analyzing generic beams and blades with arbitrary cross-sectional geometries. The code has been successfully applied to the Korean Helicopter Program development and measurement campaign of HART II blade properties, working jointly with NASA Ames, German DLR, and Konkuk University [15,16]. KSEC2D is equivalent to the Euler–Bernoulli formulation for bending and to the St. Venant level for torsion, which leads to $4 \times 4$ stiffness constants. KSEC2D II follows Timoshenko's method for bending and shear, and Vlasov's expression for torsion and shear, resulting in $9 \times 9$ stiffness coefficients. Either code can describe the effects of the elastic couplings induced due to nonzero fiber orientation angles of composites in the framework of anisotropic elasticity theory, though a more rigorous representation for nonuniform torsion and shear is implemented in KSEC2D II. A brief description of KSEC2D II is presented below.

The beam formulation is established based on the displacement-based anisotropic elasticity theory, which incorporates the influence of nonuniform warping in torsion and shear. The kinematic relations for the sectional warping displacements are derived from the energy equilibrium valid for static loading conditions. The principle of virtual work is applied to derive the governing equations of the beam. The resulting stiffness matrix can be written in symbolic form as:

$$\mathbf{K} = \begin{bmatrix} \mathbf{K}_T & \mathbf{K}_{TW} \\ \mathbf{K}_{TW}^T & \mathbf{K}_W \end{bmatrix} \tag{1}$$

where $\mathbf{K}_T$ is a $6 \times 6$ Timoshenko-like stiffness matrix, $\mathbf{K}_W$ is a $3 \times 3$ direct warping stiffness matrix due to nonuniform shear and torsion, and $\mathbf{K}_{TW}$ describes coupling between generalized Timoshenko and section warping, respectively. As a generic beam analysis code, the section properties are provided at user-defined axes (e.g., shear center, tension center). Several finite element libraries, such as 3- and 6-noded triangular elements as well as 4- and 8-noded quadrilateral elements, are provided to model arbitrary profiles (thin-walled or solid) of blade sections. The recovery of sectional strains and stresses under various loading conditions is also incorporated into the composite beam analysis. More details on theoretical aspects and validation examples can be found in [19,20].

### 2.4. Modeling Sensor and Actuator Cables

As discussed previously, the blade sections (particularly in the inboard regions) contain a number of sensor cables and actuator wires. The diameter of each cable is no more than 0.4 mm (sensor cables 0.15 mm, actuator wires 0.4 mm), including the Teflon® coating. Their impact on the blade stiffness may be negligible, but a substantial influence is expected on the mass and inertia properties due to the large number of wires and high density of the copper material in use. To avoid over-complexity in the blade cross-sectional analysis while retaining computational efficiency with less modeling effort, the cables are considered

separately from the other load-bearing structural elements of the blade. An equivalent mass lumping technique is introduced for this purpose. The total mass (lumped mass) and the associated CG locations of the cables (more exactly, groups of sub-cables) are obtained by summing up the individual elements, as given by:

$$m^c_{\text{total}} = \sum_{i=1}^{N_c} m_i \tag{2a}$$

$$y^c_{CG} = \sum_{i=1}^{N} \frac{m_i y_i}{m_i}, \quad z^c_{CG} = \sum_{i=1}^{N} \frac{m_i z_i}{m_i} \tag{2b}$$

where $N$ is the total number of cables in a group, the superscript $c$ denotes the cable, and $(y_i, z_i)$ correspond to the coordinates of the $i$-th cable element in the blade section. The corresponding mass moments of inertia (MOI) about the respective CG of the cable groups $(I^c_{y_{CG}}, I^c_{z_{CG}})$ are evaluated by applying the parallel axis theorem while summing the individual contributions of the cables:

$$I^c_{y_{CG}} = \sum_{i=1}^{N} m_i z_i^2, \quad I^c_{z_{CG}} = \sum_{i=1}^{N} m_i y_i^2 \tag{3}$$

The mass lumping technique is illustrated in Figure 7 for section U6, stationed at 0.9125$R$ of the blade. Figure 7a shows a fully developed FE mesh at this section, in which all cables and structural elements are discretized accordingly. The FE modeling was performed using the commercial pre- and post-processor software MSC.PATRAN. A 4-node quadrilateral element (QUAD4) was employed for the spar and skin, whereas in the cable domain and the material form domain, a 3-node triangular element (TRI3) was used. A total of 46,025 elements were included in the model. Due to the irregular shapes and narrow corners near the zones surrounding the cables, very fine meshes were inevitably formed, as can be seen in Figure 7a,b, which shows the FE mesh using the mass lumping technique described above. The rectangles in red, green, and blue denote the respective lumped elements that counter the individual mass and inertia of the pressure cables, strain gage cables, and actuator cables, respectively. The number of FE mesh elements is 6578, which corresponds to about 15% of the full FE model.

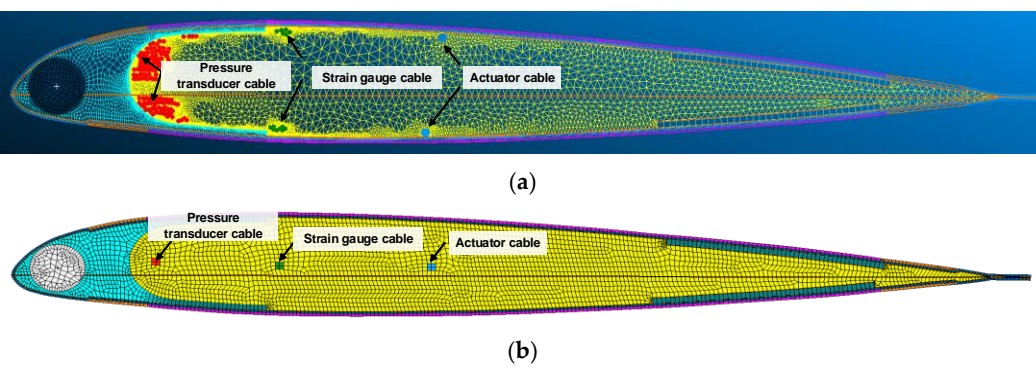

**Figure 7.** FE mesh of a blade airfoil section (U6) incorporating sensor cables. (**a**) Full FE mesh with individual cables. (**b**) Reduced FE mesh with lumped mass of cables.

The accuracy of the mass lumping technique was assessed in terms of the direct FE analysis predictions. Figure 8 compares the predicted values for the section offsets, mass (mass per unit length), and inertia (MOI) of section U6 between the mass lumping scheme and the full FE model presented in Figure 7. The centroid offsets (shear center, tension center, and CG) of the section were evaluated in terms of the length (mm) from the LE. As can be seen, an excellent correlation is obtained between the two sets of predictions. The maximum discrepancy is within 1% in reference to the baseline values of the full FE model,

demonstrating the accuracy of the mass lumping method when modeling the cables. Based on the correlation results, the mass lumping technique was adopted in the subsequent blade cross-section analysis.

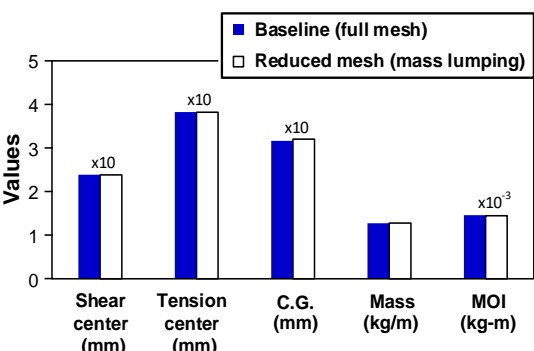

**Figure 8.** Effect of cable modeling on section offsets, mass, and inertia properties at U6.

### 2.5. Composition of FE Cross-Sectional Analysis Model

The 2D FE discretization process was performed based on the CT scan images combined with the digital segmentation task described earlier. Figure 9 shows the FE meshes discretized using 8-node quadrilateral elements (QUAD8) for some of the cross-sections in the blade root region (R2, R5) and blade airfoil region (U2, U7). The number of elements used to discretize the sections are provided in the plots. Generally, 4700–7500 elements were required to model the sections (all 14 stations), except for the innermost sections at the blade root region (R1–R3). The reason for the large numbers of elements in the inboard root regions is the discretization required to cover the thickness of each constituent layer of the spar structure, which consists of alternating fiber angles of ($0°/90°/\pm 45°$). Up to 274 layers of UD CFRP are used to fabricate the laminated spar in the blade root. Note that the computational burden of modeling and analyzing the sections is significantly decreased by applying the mass lumping technique.

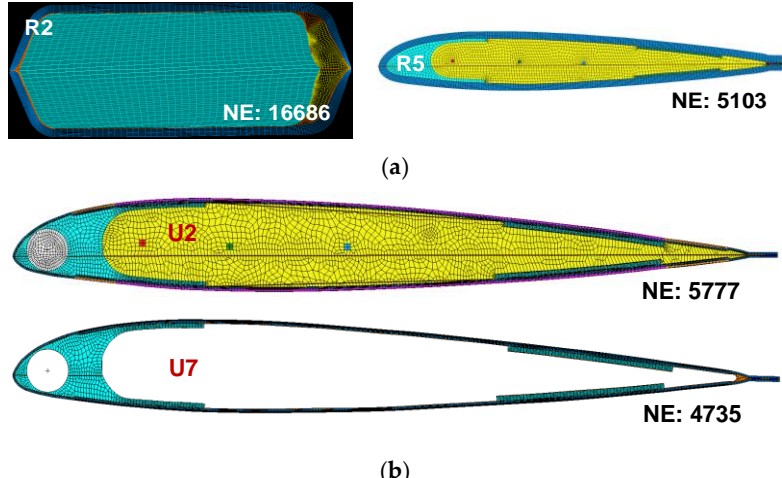

**Figure 9.** FE meshes at the selected sections of the blade (NE: number of elements). (**a**) Blade root sections and (**b**) blade airfoil sections.

Figure 10 summarizes the overall flow of the generation of 2D cross-sectional analysis models of STAR blades. The work flow begins with the X-ray CT scans, followed by the section segmentation to identify subregions with the same material domains over the sections. The segmented section images are imported into a CAD software program (CATIA) that converts the images into a neutral file format (e.g., IGES files). A mesh seed on the interior of the geometric surfaces is formed based on the graphical data with the aid

of a commercial pre-processor software such as MSC.PATRAN. Next, the lumped mass technique is applied to the cables to determine the equivalent positions of their concentrated masses over the sections. FE discretization is then applied to the cross-sections prescribed by the domain boundary surfaces. Finally, 2D blade cross-section analysis is conducted to obtain the structural properties of the designated sections.

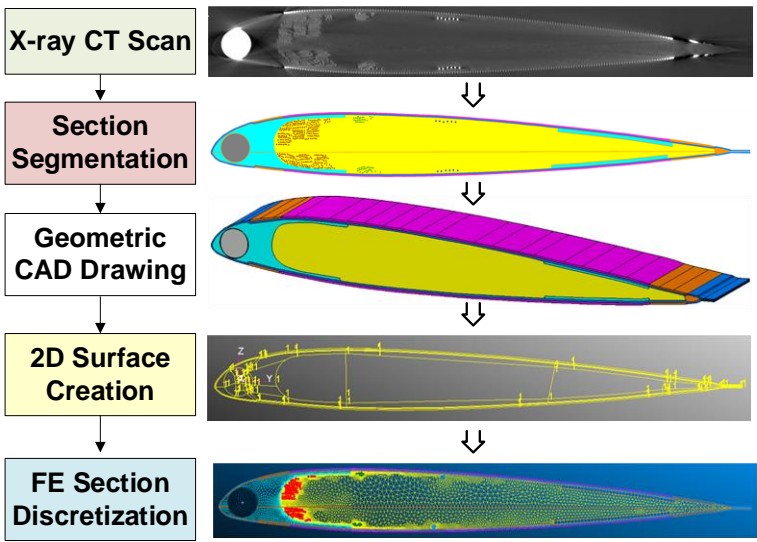

**Figure 10.** Flow diagram for building the FE blade section analysis model.

## 3. Measurement of Blade Structural Properties

A mechanical (non-destructive) measurement technique was employed to evaluate some of the blade structural properties, confirm the fidelity of the constructed analytical model, and assess the correlation with the predicted results. The test rig used for HART II blades [16] was updated and used in the present study. Figure 11 shows the measurement setup for a STAR II (epsilon) blade. Only the uniform (in an aerodynamic sense) portion of the blade was considered in the measurements. The blade was clamped at an inboard root location (0.24*R*) and loaded at the tip through a wire-pulley system attached to a loading lever (see Figure 11). The effective blade span length is 1525 mm from the root fixture to the point of load application. The position of loading can be varied arbitrarily along the chord direction using a sliding mechanism attached to the loading lever. The force was applied in the direction perpendicular to the blade axis. The load was applied using a planar disk rotated at 0.3 Hz and connected to the wire-pulley system. This quasi-static loading condition was used to avoid the creeping effects of the glass fiber materials of composite blades [22,23]. The tip deformations were measured using a high-resolution laser profile scanner (Micro-Epsilon 2910-100) installed on the test rig near the blade tip. The measured deflection data were processed (for linear displacements and angles) in real time on a Windows PC using the software provided by the manufacturer.

The location of the elastic axis was first identified. For this purpose, the external load was initially applied at the quarter chord position, and then shifted towards the LE or TE along the chord using the metal lever mentioned above. This loading was repeated 5–10 times and the deformation data were averaged to mitigate possible measurement errors. The resulting section rotation angles were monitored with respect to the points of loading.

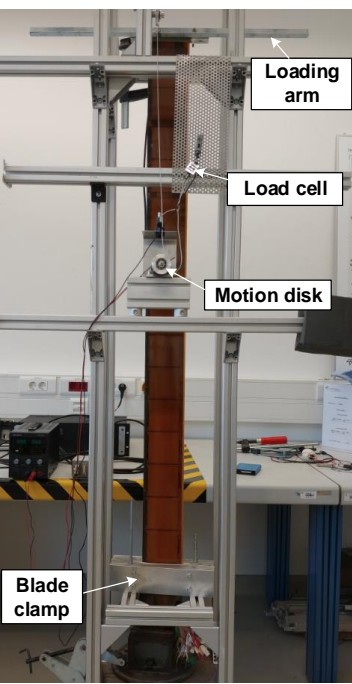

**Figure 11.** STAR blade in the lab bench test for elastic property evaluation.

Then, they were fitted to a curve to determine the position of zero section rotation, which essentially corresponds to the elastic axis location. Note that the elastic axis is independent of the magnitude of the loading and is defined as the chordwise position at which the beam bending introduces no torsional motion. Once the elastic axis location had been found, the quasi-static loads were applied at the blade tip to determine the blade bending or torsional rigidities. The flap bending stiffness, $EI_y$, and torsional stiffness, $GJ$, are evaluated by the relations:

$$EI_y = \frac{F \cdot l_e^3}{3 \cdot \delta_{tip}} \tag{4a}$$

$$GJ = T \cdot \left( \frac{d\phi}{dx} \right)^{-1} \tag{4b}$$

where $F$ and $T$ are the external force and applied torque respectively, and $l_e$ is the effective beam length, $\delta_{tip}$ is the elastic bending displacement, and $\phi$ is the twist angle. When obtaining the section stiffness values, five different measurements were performed and averaged while varying the amplitudes of the applied loads in each test, and this was enabled by adjusting the lengths of the loading wires (see Figure 11). Dynamic loads of 7–8 N (peak-to-peak values) were applied.

## 4. Results and Discussion

The fidelity of the blade analytical model constructed using the proposed digital reproduction process is first assessed by comparing the estimated blade structural weight with the measured weight data. Next, 2D sectional properties at specified radial stations are evaluated through comparisons with the measured data. Finally, a sensitivity analysis is carried out to examine the impact of modeling non-structural elements (e.g., cables, nose weight) on the blade structural property evaluations. The effects of manufacturing imperfections are also examined using the TE spar structures.

### 4.1. Estimation of Blade Structure and Cable Weight

The weight of blade structures and wiring cables was estimated based on the geometric and material properties of the digital blade model constructed as described in the previous sections. Note that the influence of cables on the blade weight is substantial because of the

large number of sensors and actuators (as many as 553) distributed within the blades, as well as the high density of copper material in the cables. Figure 12 shows a schematic of a cable cross-section consisting of copper-silver wire (with a circle of radius $r_1$) enclosed by Teflon® coating ($r_2$). It is assumed that a thin layer of epoxy resin ($t_{resin}$) is impregnated uniformly over the Teflon® surface. The cables must be tightly glued to fix their positions in the space of the section during the entire service life while enduring the centrifugal action of the rotor. The respective dimensions of the sensor and actuator cables used for the fabrication of the blades are summarized in Table 2.

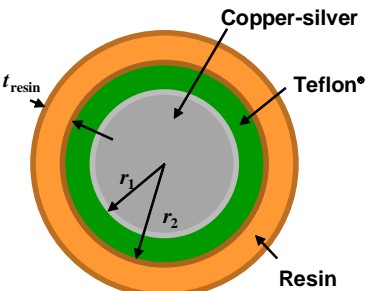

**Figure 12.** Composition and geometry of cable sections.

**Table 2.** Geometric dimensions of the cables.

| Material | $r_1$ (mm) | $r_2$ (mm) | $t_{resin}$ (mm) |
|---|---|---|---|
| Sensor cables | 0.076 | 0.152 | 0.1 |
| Actuator cables | 0.152 | 0.402 | 0.1 |

The total mass of the blade structures and the collected cables was estimated using the scan images (Figures 3 and 4) and the projected areas (Figure 7) computed using KSEC2D II with the help of CAD software. It is emphasized that the spanwise stations selected for the section discretization (R1–R7, U1–U7) were determined carefully based on the principle that substantial variations exist in either the exterior or interior layouts of the blade images. The exterior variations are governed by the geometric shapes (curvatures and actuator edges), while the interior variations are dominated by the location of the pressure sensors and the associated cable harness connected to them. This confirms that an almost-uniform distribution of cables can be assessed for a volume segment bounded by the nearby stations of the blade. Figure 13 shows the volume representation of the blade root region composed of R0 (identical to R1) to U1, along with a planar view of the blade airfoil region covering U1 to the blade tip. Regarding the blade root region, the depth and width of which vary along the span, each volume of the corresponding segments was computed using the outer mold line (OML) curves prescribed by CAD software. In the airfoil blade region, which is regarded as uniform along the blade length, each volume segment is represented using a pultrusion process based on any section, $U_i$ ($i = 1$–7), prescribed by the airfoil contour (NACA 23012). The volume of the constituent structural elements (e.g., skin, spar) at each segment of the blade was determined by interpolating the surfaces of the material domains enclosed by the two adjacent sections ($i$, $i + 1$) of the blade, as depicted in Figure 13a. Multiplying the material density of the domain and summing over the volume segment, $V_i$, yields the mass of a segment. The total mass of the blade structure (except the cables) was obtained by summing the individual components over the segments. The mass of the cables at blade volume segment $i$ is calculated as follows:

$$m_{cable}^{(i)} = \sum_{k}^{3} \left\{ \frac{\left( A_i^k + A_{i+1}^k \right) l_i}{2} \cdot \rho^k \right\}$$

(5)

where the superscript $k$ denotes the type of the cables ($k = 3$, two sensors and one actuator), $A_i^k$ is the cross-sectional area of the cables at the $i$-th section, $l_i$ is the segment length, and $\rho$ is the material density. Note that the mass of the exposed cables near the blade root (see Figure 1) was estimated to be approximately 190 g.

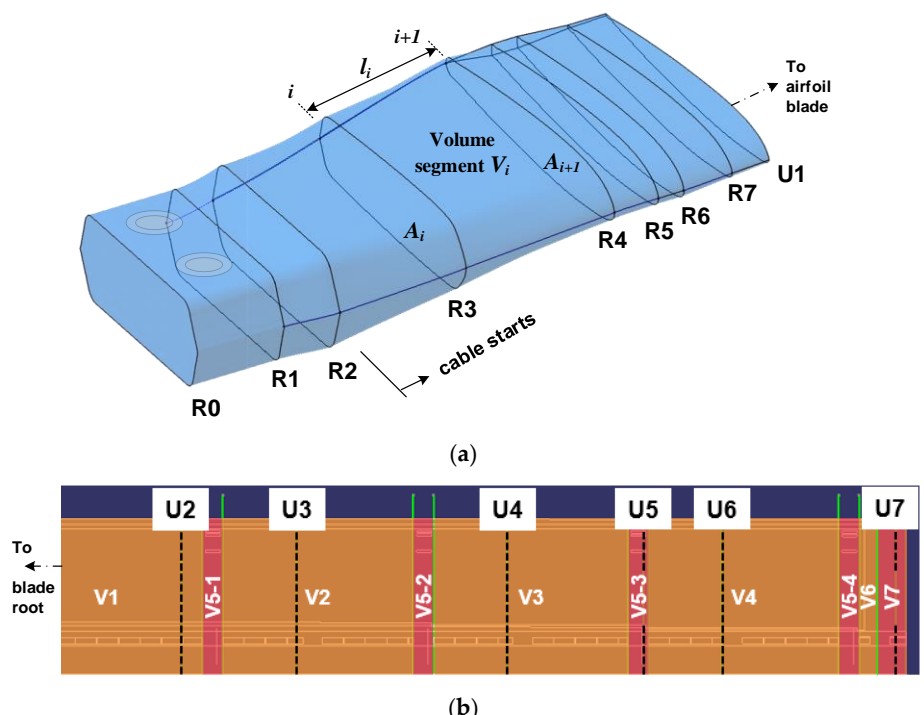

(**a**)

(**b**)

**Figure 13.** Subdivision of blade segments for estimating interior volumes. (**a**) 3D layout for blade root region. (**b**) Planar view of blade airfoil region (inboard region not shown).

Figure 14 shows the correlation between the predicted blade mass estimated using the method described above and the measured data. The measured blade weight including the cables was close to 3 kg. The predicted result is in excellent agreement with the measured data, with a slight underprediction. The cables were estimated to weigh about 0.459 kg, while the pure structural weight of the blade was found to be 2.528 kg. As the fabrication work was performed manually, there remains a certain level of uncertainty in the resin content surrounding the cables. Other manufacturing uncertainties might exist in the blade construction process. Taking some of these uncertainties into account, the predicted mass appears to capture most of the blade structural elements reasonably well. This relatively good correlation indicates the high fidelity of the digitally reconstructed model of the blade.

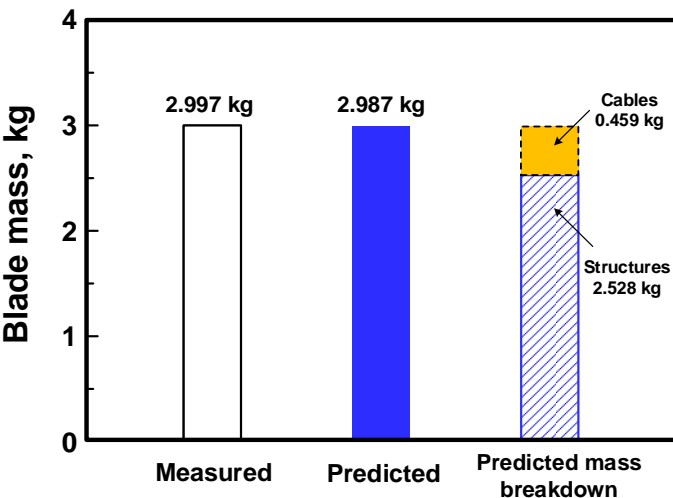

**Figure 14.** Validation of predicted mass against measured data.

### 4.2. Blade Structural Properties

Blade structural properties such as the centroidal offsets, stiffnesses, and sectional mass and inertia have been evaluated. Figure 15 compares the section centroidal offsets obtained for the blade airfoil region from the mechanical measurements and 2D FE cross-sectional analysis using KSEC2D II. The predicted results were obtained by averaging the predictions of each blade airfoil station (U1–U7) to ensure a fair comparison, despite minor deviations between the different sections. Both flap-up and flap-down measurements were taken to determine the measured elastic axis (shear center) position. The measurements show substantial variations between the flap-up and flap-down configurations, and so the values were averaged to obtain the final measured data. The reason for these deviations is unclear at this time, although geometric pre-twist angles of $-8°$ and the MFC patches (positioned at $\pm45°$ with respect to the beam axis) installed on the top and bottom surfaces may have an effect.

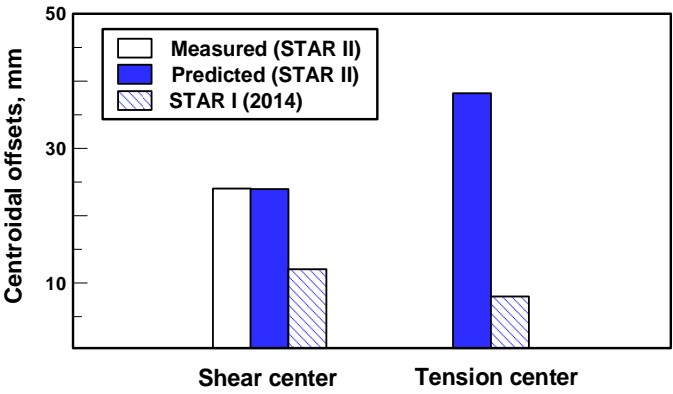

**Figure 15.** Comparison of centroidal offsets.

The correlation between the predicted results and the measured data for the STAR II blades is excellent. Both methods predict the elastic axis location to be close to 24 mm from the LE (20% chord). In Figure 15, the earlier STAR I results [11,12] are presented for reference. Both the shear center and the tension center (modulus-weighted centroid) have shifted significantly toward the TE owing to the modifications implemented in the STAR II blades. The large shift in the tension center (at about 25% chord) toward the airfoil TE can be attributed to the addition of TE spars.

The spanwise distribution of the structural properties of STAR II blades is presented in Figure 16 alongside measured data and the results from HART II blades [16]. The properties have been nondimensionalized by the corresponding (representative) values at the blade

airfoil region to enable a clearer comparison. The stiffness properties were only measured at the blade airfoil region, and they are presented together in Figure 16. The agreement is reasonable (maximum error of within 8.1%) for the bending and torsional rigidities. The STAR II blades exhibited significantly higher variations along the blade length than HART II, and this is due to the fundamental differences between the rotors. For example, the articulated configuration of the STAR II rotor demands much higher bending stiffness over the blade root region than the hingeless rotor in HART II. The large scatter in the distribution of sensors and the related cable harness also increase the variations between the blade stations (i.e., changes in cables for U1–U7, pressure sensors for U5, and material composition for U7). In the HART II blades, the properties are uniform throughout the blade airfoil region [16].

Figure 16 indicates that the maximum variation in the flap bending stiffness of HART II blades was about 3.5 times based on the blade airfoil region, whereas the STAR II rotor had a variation of 44 times along the blade span length.

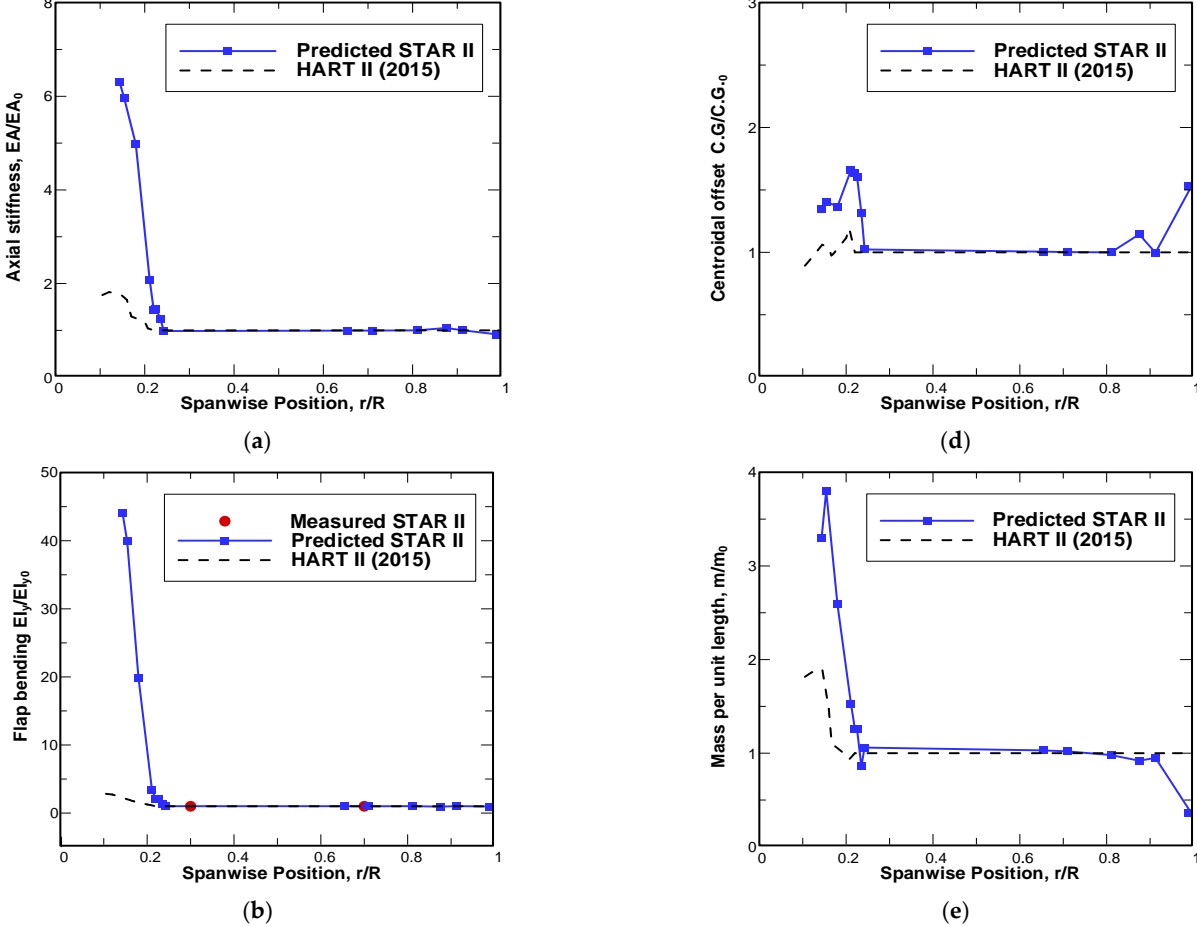

**Figure 16.** *Cont.*

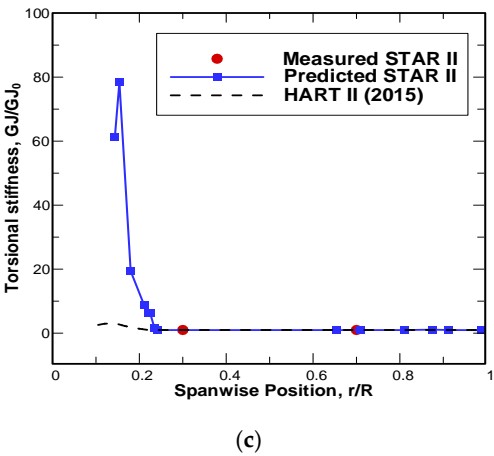
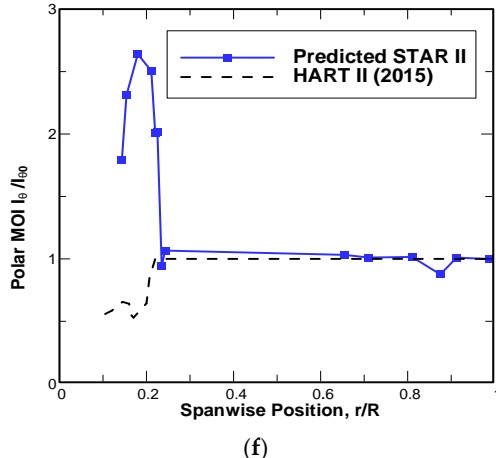

(**c**)  (**f**)

**Figure 16.** Comparison of blade structural properties (nondimensionalized by their uniform values). (**a**) Axial rigidity, (**b**) flap bending rigidity, (**c**) torsional rigidity, (**d**) section CG offsets, (**e**) mass per unit length, and (**f**) polar MOI.

Specifically, at the blade root station ($x/R = 0.145$), the flap bending stiffness of STAR II was 35 times higher than that of HART II. Due to the high stiffness of the STAR II blades, especially in the blade root region, the test setup demands a very high level of loading to achieve reasonable measurement data. The test setup shown in Figure 11 has been successfully applied to HART II blades, covering both the blade airfoil region and the blade root region. The amount of loading should be too small to evaluate the stiffness properties in the root region of STAR II blades under the current test setup. Furthermore, considering the high cost of the MFC patches distributed over the blade surface (15 patches per side), there is a clear concern about the application of large loadings for structural property measurements. Hence, only the blade uniform region with limited loadings was considered for the measurement activity.

*4.3. Model Sensitivity Analysis*

Finally, the results of a sensitivity analysis to identify the influence of specific structural (or nonstructural) elements and manufacturing imperfections on the blade structural properties are presented. The design parameters considered include the sensor and actuator cables, nose weight, and imperfections in TE spar construction for the cross-section analysis, as depicted in Figure 17. For the former two parameters, each constituent part (area) was discretized and allocated the corresponding material properties (including cables), assuming that they were serving as ordinary structural elements in the blade composition. Counterpart models that neglect the structural stiffness contribution were built as a reference. In this way, the effect of incorporating cables or nose weight on the structural properties can be examined in a quantitative manner. The final parameter chosen was associated with manufacturing imperfections (misfits) found in the CT scan images (see Figure 17). For the sensitivity study, an FE cross-sectional model with adjusted (vertically aligned) fits for the TE spar was generated and compared with the original misfitted model.

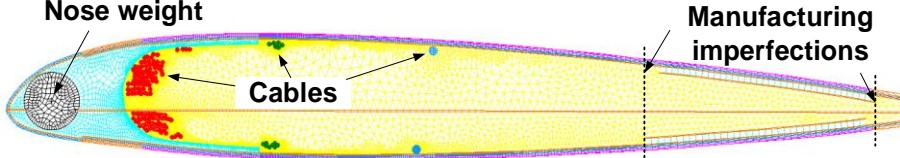

**Figure 17.** Design parameters used for the sensitivity study of blade section U6 (0.9125*R*).

Figure 18 shows the sensitivity analysis results for the blade structural properties with respect to the three parameters described above. The percentage differences (absolute scale) in the parameter values refer to those obtained when their structural contributions

were neglected. Section U6 was selected because it contains relatively few cables. Sections further inboard require more involved analysis, as they contain a larger number of cables (Figures 4 and 6). As can be seen in Figure 18, the influence of neglecting each component was non-negligible. When the cables were neglected, the mass changed by about 4.3%. The effect on CG and MOI was limited (within 1.5%) because most of the cables are already assembled near the chordwise CG location. As expected, the effect of the cables on flap bending and torsional stiffness was negligible. Therefore, it is reasonable to assume that the cables affect only the mass and inertia properties.

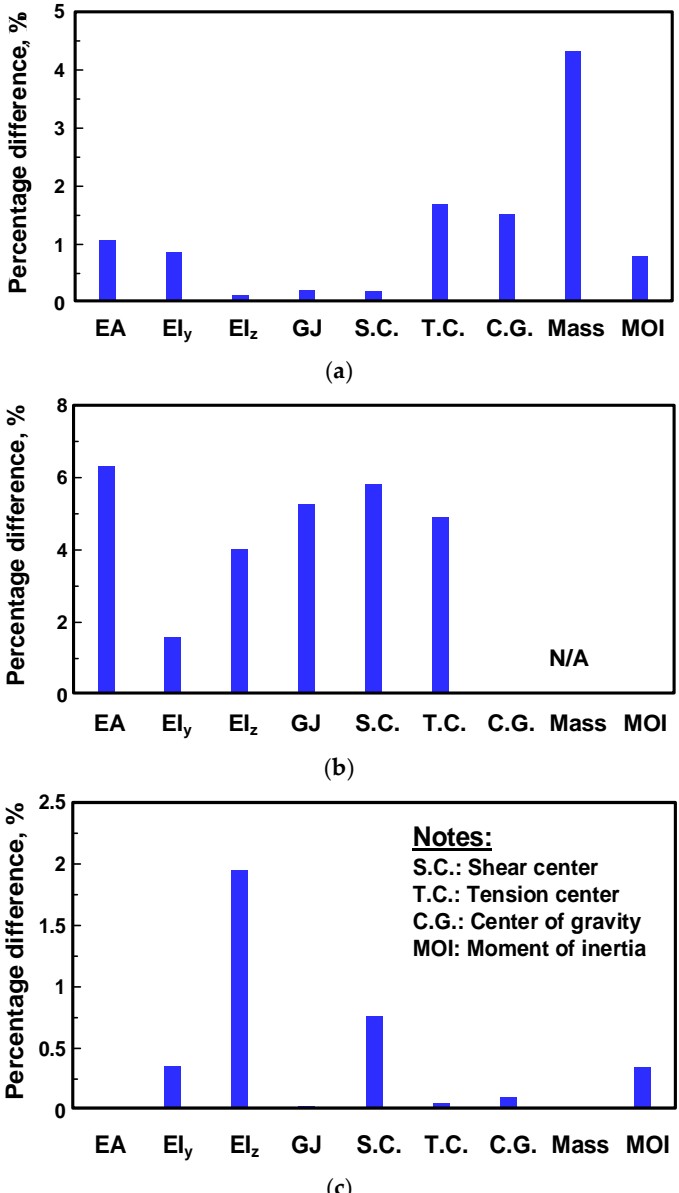

**Figure 18.** Sensitivity results on structural properties of the design parameters. (**a**) Effect of cable modeling, (**b**) effect of nose weight modeling, and (**c**) effect of imperfections on TE spar modeling.

The influence of incorporating a nose weight on the evaluation of structural properties was significant (see Figure 18b), and should thus be considered carefully. The percentage error from neglecting the elastic property reached 6.3%, and mainly affected the axial rigidity and the elastic axis. The reason for this large degree of influence is the use of titanium in the present model. Some nose weight materials (e.g., lead) only affected the mass and inertia characteristics, and their contribution to the stiffness and centroidal

offsets can be neglected. Based on this observation, accurate modeling of the nose weight appears to be crucial in the structural property evaluations and associated structural dynamics analysis.

Figure 18c shows the effect of manufacturing imperfections induced through misalignment of the TE spar (top and bottom laminates) on the structural properties. The TE spar was fabricated using five layers of UD CFRP, each with a thickness of 0.125 mm. The counterpart model perfectly matched the starting locations of the top and bottom laminates, while keeping the wall thickness of the laminates identical. The influence on properties such as mass, flap bending, and torsional stiffness was seen to be marginal, as the area of the TE spar was almost the same in both models. The most significant effect was on the chord bending stiffness, because of the large offsets from the shear center. The deviation was about 1.95% compared with the perfect fit case. The influence on the shear center location appeared to be non-negligible, considering the small magnitude of the misfits in this study (about 1.5 mm).

## 5. Conclusions

In this work, the structural properties of STAR II blades were evaluated using a digitally reconstructed FE analytical model that reflects real manufactured blades. An X-ray CT scan scheme was combined with image segmentation to cover the whole blade, including the blade root, transition, airfoil, and tip regions. The detailed lamination geometries of the skin and spar structures were incorporated in the model as in the manufactured configuration. A non-destructive mechanical measurement method was used to obtain the elastic axis and the stiffness properties. The following conclusions can be drawn from this study:

(1) Digitally replicated, high-precision analytical models of a STAR II blade were obtained from the image processing framework proposed in this study. The constructed sectional FE model reflected much of the physical, fabricated STAR blades, including sensor and actuator cables, detailed lamination geometry of composites, and manufacturing imperfections.

(2) The fidelity of the digitally reconstructed model was assessed by correlating the predicted blade weight with measured data. The predictions showed an excellent agreement with the measured weights. A breakdown of the predicted weight revealed that the cables constitute 15.3% of the total blade weight.

(3) The measured elastic axis location exhibited substantial deviations between the flap-up and flap-down measurements for some unidentified reason. The averaged results between the two measurement configurations were in good agreement with the predicted position at 20.08% chord. The predicted stiffness results showed a reasonable correlation with the measured data. The discrepancy was about 8.1% with the measured flap bending. The spanwise distributions of the estimated blade properties were identified and compared with those of earlier HART II blades.

(4) The sensitivity of modeling secondary elements (e.g., sensor cables, nose weight) and misfits (manufacturing imperfections) in the TE spar on the blade structural properties was examined for a cross-section at 0.912$R$ (U6). The impact was found to be non-negligible for the parameters considered. Neglecting their respective effects in the section analysis led to a 4.3% error in mass for the cables and a 6.3% error in axial rigidity for the nose weight. The introduction of small misfits (about 1.5 mm) in the TE spar resulted in an error of 1.95% in the chord bending stiffness, which is non-negligible due to the large offset from the elastic axis.

**Supplementary Materials:** The following are available online at https://www.mdpi.com/article/10.3390/aerospace8120370/s1, Figure S1: High resolution image of Figure 3b, Figure S2: High resolution image of Figure 4a: R3, Figure S3: High resolution image of Figure 4a: R5; Figure S4: High resolution image of Figure 4a: R7; Figure S5: High resolution image of Figure 4b: U3; Figure S6: High resolution image of Figure 4b: U5; Figure S7: High resolution image of Figure 4b: U7.

**Author Contributions:** Conceptualization, S.N.J.; section segmentation, J.H.A.; FE sectional analysis, H.J.H.; visualization, S.C.; measurement, R.K.; supervision—measurement, S.K.; writing, S.N.J.; supervision, S.N.J. All authors have read and agreed to the published version of the manuscript.

**Funding:** This work was supported by the National Research Foundation of Korea (NRF) grant funded by the Korea Government (MSIT) (NRF2021R1F1A104557011). This work was conducted at the High-Speed Compound Unmanned Rotorcraft (HCUR) research laboratory with the support of the Agency for Defense Development (ADD). This paper was supported by the Konkuk University Researcher Fund in 2020.

**Institutional Review Board Statement:** Not applicable.

**Data Availability Statement:** The data presented in this study are available upon request from the corresponding author.

**Acknowledgments:** The authors would like to acknowledge the technical support of Thomas Ullmann at DLR Stuttgart and Bram van de Kamp at DLR Braunschweig in the use of X-ray CT test facility.

**Conflicts of Interest:** The authors declare no conflict of interest.

## Nomenclature

| | |
|---|---|
| A | Cross-sectional area |
| $E_L$, $E_T$, $G_{LT}$ | Elastic moduli (E, G) in longitudinal (L) and transverse (T) directions |
| EA | Axial stiffness |
| $EI_y$, $EI_z$ | Blade bending stiffness in y and z directions |
| GJ | Torsional stiffness |
| $I_y$, $I_z$ | Sectional MOI in y and z coordinates |
| l | Blade length |
| m | Mass |
| R | Rotor radius |
| t | Wall thickness |
| $y_i$, $z_i$ | Cross-sectional coordinates |
| $\delta$ | Blade displacement |
| $\phi$ | Elastic twist angle |
| $\rho$ | Material density |

## Abbreviations/Acronyms

The following abbreviations and/or acronyms were used in this manuscript:

| | |
|---|---|
| ATR | Active Twist Rotor |
| CAD | Computer-Aided Design |
| CFRP | Carbon Fiber-Reinforced Composites |
| CG | Center of Gravity or Mass Center |
| CT | Computed Tomography |
| DLR | German Aerospace Center |
| DNW | German–Dutch Wind Tunnel |
| FE | Finite Element |
| HART | Higher-harmonic Aeroacoustic Rotor Test |
| IGES | Initial Graphics Exchange Specification |
| LE | Leading Edge |
| MFC | Macro Fiber Composite |
| MOI | Moment of Inertia |
| OML | Outer Mold Line |
| RGB | Red Green Blue |
| SC | Shear Center (or Elastic Axis) |
| STAR | Smart Twisting Active Rotor |
| TC | Tension Center |
| TE | Trailing Edge |
| UD | Unidirectional |

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
