# Peer review of "X-ray Computed Tomography Method for Macroscopic Structural Property Evaluation of Active Twist Composite Blades"

_aerospace, doi:10.3390/aerospace8120370_

Round 1

Reviewer 1 Report

The paper entitled “X-Ray Computed Tomography Method for Macroscopic Structural Property Evaluation of Active Twist Composite Blades” focuses on evaluating the structural properties of helicopter blades without destroying them. As a solution, it presents how X ray computed tomography can be used as an input for generating structurally-faithful finite element models of the next-generation active twist blade for helicopter applications.

The article is divided into 5 sections. It starts with an introduction of the vibration problems in helicopters and of the active twist rotor (ATR) scheme as a potential solution. The second section presents how X ray computed tomography can be used to obtain the structure of the blade and separate its different parts, then produce a 2D finite element model that is faithful to the actual structure of the physical blade. The third section deals with the non-destructive measurements of the physical blade, used to validate the FE models generated from CT data.  The fourth part gathers and discuss the results in terms of estimation of blade structure and cable weight, blade structural properties, and model sensitivity analysis. The fifth part is the conclusion of the article.

Overall, the article is well written and clear, clear enough for a non-expert in aeronautics to understand the significance for the field of the results being presented. From an X-ray computed tomography point of view, micro-structurally faithful models are not new, but the article is very detailed in the assessment of the results obtained by the FE modelling, and their comparison to the non-destructive assessment of the physical blade which provide significant improvements to the field of aeronautics. I also found the results to be very complete and the model sensitivity analysis to be very interesting. I recommend this article for publication with minor revision, my comments being included below. My comments cover mainly the X-ray computed tomography parts (the introduction and section 2). Although I have also reviewed parts 3 to 5 and did not detect any major problems, I do not believe, however, that my level of expertise allows me to provide significant feedback on these parts.

Author Response

Page 2 line 66: “structure is that each voxel (volume pixel) image has a discrete CT number (Hounsfield unit)” voxel stands for “volume element”, not “volume pixel”, but it is true that it is the 3D counterpart to a2D pixel

Ans) Thanks for the comment. The wording has been modified as suggested.

In addition, the Hounsfield unit is mainly used in the medical field. In material science, there is no unit, each voxel has a given grey scale value which is proportional to the density of the material over the voxel volume within the sample

Ans) The wording has been modified as: “(called Hounsfield unit in the medical field)” reflecting the comment.

Page 3 line 124 “pixel pitch” instead of “pixel size”. Pixel pitch is used to differentiate the physical size of the pixel on the 2D area detector to the size of the pixel in a 2D projection. The former is constant whilst the later changes with the magnification employed.

Ans) The wording has been modified as suggested. Thanks for the correction.

Page 3 line 125 the term “tomogram” is incorrect as it refers to reconstructed data. A 2D tomogram usually refers to a slice from the fully reconstructed volume (which is the tomogram). Here the more appropriate term should be “2D projection image” as it is the raw data recorded during acquisition.

Ans) The wording has been modified as suggested. Thanks for the suggestion.

Page 3 line 127 “fan beam shaped” is incorrect, strictly speaking the X ray source emits x-rays nearly all direction, but the detector defines the geometry of the CT system. Here, as the detector is a flat panel, the geometry is referred to as “cone beam”. “Fan beam” would be in the case of a curved or flat linear detector array.

Ans) The wording has been modified as suggested. The authors thank for the correction.

The current of the x ray gun and pixel/voxel size are missing in the description of the CT scanning (section 2.2). Those are important parameters for experiments to be duplicated. In addition, the power of the X ray gun and the voxel size are important to compare in order to get an idea of the possible blurring in the images. Both current (or power, as the voltage is given) and the voxel size need to be added before publication.

Ans) The technical data for the power (1500 Watt) and the voltage (450 kV) of X-ray tube were included in the revised manuscript. The voltage data given in the earlier version was misled, so the related sentence was removed. The volume pixel size used for the scan (361 mm) was specified in the section 2.2.

Page 4 line 157: “The respective zones inside the blade structure are identified in Figure 3(b)”, the grey scale value across scans S1 and S2 are not consistent thus, the same material in each of the volume has a different grey scale (particularly visible on the LE Spar CFRP part).

Ans) We fully agreed with the reviewer’s opinion. The grey scales in each scan should not be the same due to different material compositions. None the less, what we meant was that the local zones with different materials were identified considering the grey scale images with the aid of the known topology of the blade structure.

Scales on the images are missing on Fig 3 and 4

Ans) Though not clearly seen, the scale is already presented in Figure 3. This (20 mm range) can be seen in the mid of the left-hand edge of the figure. Because of the similarity in the background color, there is a difficulty in identifying the scale. It will be identified if the figure is magnified. The other CT images have their own scales which were placed somewhere in the bottom corner of the images, and they were intentionally removed while cutting them in order to reduce the size. This is unavoidable since there is a limitation (50 MB) in the file size allowed to upload on the journal review system.

Page 4 line 187 please name the software (and version) used

Ans) VGSTUDIO version 3.2.0 was used to visualize the CT data. The software name with the version was included in the text.

Page 6 line 196: “It should be emphasized that the manufacturing defects or imperfection zones (except artifacts) found in the CT images …” there is no mention of the dark horizontal line in the centre of the CFRP spar and also from the 2 brighter lines that can be seen in Fig 5a in the segmented image. Similar remark for Fig 6, as the CT data are quite noisy, how accurately were determined the thicknesses of the GFRP skin and MFC actuator. The resin is also extremely difficult to visualise on the CT image.

Ans) It is acknowledged that there have been serious disputes about the way how and whether to model the lines that were found in the root sections (R1 to R3), as pointed out by the reviewer. The darker line that could be seen in the center of Figure (5a) was formed during the merging stage of the individual piece (detached from aluminum molds) of the upper and lower parts of the blade. We know the material used for the fabrication is the epoxy resin (relatively soft material). Whereas the two bright lines are unidentified and still remained unknown until at this moment. Neglecting those lines in FE model might result in some tiny errors in estimating the section structural properties, due to a small thickness in the resin layer (for the darker line) and similarity in elastic properties between GFRP and CFRP composites (for the white lines). Considering those thin lines in FE model will help increasing the complexity of the model with a drastic increase in the number of finite elements. Therefore, we decided not to incorporate them in the analytical model. It is believed that the error range due to the neglect of the thin lines will certainly be very low (maybe out of the effective digits). In the revised manuscript, the following statements are added: “It is noted that the dark horizontal line and the brighter lines captured in Figure (5a) are neglected in the segmentation stage considering the feature that their impact on the structural properties should be marginal due to a small thickness in the resin layer and/or similarity in elastic properties between GFRP and CFRP composites. The incorporation of the thin layers in the model may contribute to a drastic increase in the number of degrees of freedom (number of FE’s) with no significant changes.” Regarding the second comment, it is true that the thicknesses of the GFRP skin and MFC actuator are hardly differentiated (due to a strong illumination by MFC actuator materials) while the resin layers are identified by magnifying the images and/or comparing the nearby section CT images. In the former case, both CAD drawings (i.e., known topology) and manufacture data are used to determine the boundary between the two material zones. This has been stated in the manuscript as: “The respective zones inside the blade structure are identified…, taking into consideration the gray scale level and the known topology of the blade structure.”

Figure 10: use the term “segmentation” instead of “colour segmentation” throughout the document.

Ans) The wordings in Figure 10 and the text have been changed as “section segmentation”.

Page 17 line 529 “Digitally replicated, high-precision analytical models of a STAR II blade were obtained from the image processing framework proposed in this study”, I believe the voxel size is needed to understand the extend of the term “high precision” and also assess the size of the smallest defect detectable.

Ans) The voxel size data are provided in the section 2.2.

Reviewer 2 Report

This article brings a valuable contribution on the advanced performance assessment of complex industrial aeronautical parts, namely active twist composite blades. It takes into account the peculiarities of real post-manufactured blades (e.g. heavy instrumentation with sensors and actuators cables, manufacturing defects and imperfections …), which is particularly interesting and novel.

The scope is clearly defined and justified with regard to the co-authors’ previous work. The article is well-structured. The methodology and hypothesis are comprehensively described. The results are well illustrated and relevantly discussed. The conclusions are consistent and point out the major achievements.

Only downside: The rate of self-citations is huge (14/23, i.e. more than 60% of the references) and most (75%) of the references mentioned were published before 2017 (only 25% published within the last 5 years).

Thus, regarding the article type, it would be more appropriate to label this paper as “Case Study” or “Technical Note” rather than “Original Research Article”.

In addition, the following revisions are suggested:

Abstract: Add a quantification where applicable (for example, give an idea of what “good” or “excellent” means, e.g. “discrepancy/error/difference < xx%”)

Before L26: Include a list (or table) of all abbreviations/acronyms and symbols used in the text and on the figures at the beginning of the document, just before the introduction. This would be helpful to the reader.

L56: after “glass fiber composite (GFRP) ”. Add “(GFRP)”

L59: “mass center (or center of gravity, CG). Add “center of gravity” into the brackets

L119: indicate the references (supplier …) of the CT scan machine

L172:”with a polytetrafluorethylene (PTFE, Teflon®) coating”. Indicate material type (Teflon is a commercial name)

L172, 195, 246, 369, 370 and L375 (fig.12): Teflon is a trade mark. Write “Teflon®”

Table 1: “Resin” and “Foam”. Indicate the types of resin (epoxy resin ?) and of foam materials

L209 (Table1, note a,b,c): “Elastic moduli in tension (E) and shear (G) in fiber (F) and transverse (T) directions”. Add the meaning of E, G, L, T into brackets

L213 (sub-title 2.3.2): “2D cross-sectional”.

Figure 7: Replace the trade mark “Kulite” by “Pressure transducer”

Figure 18: Make sure that the notion EA (axial stiffness) is defined somewhere in the text (or add the definition to the note defining the abbreviations SC, TC, CG, MOI on fig 18c)

Author Response

Only downside: The rate of self-citations is huge (14/23, i.e. more than 60% of the references) and most (75%) of the references mentioned were published before 2017 (only 25% published within the last 5 years). Thus, regarding the article type, it would be more appropriate to label this paper as “Case Study” or “Technical Note” rather than “Original Research Article”.

Ans) The authors have tried our best to survey the up-to-date journal articles and conference proceeding papers while referring the related publications. It is hoped that the reference section contains the latest information in appropriate form.

Abstract: Add a quantification where applicable (for example, give an idea of what “good” or “excellent” means, e.g. “discrepancy/error/difference < xx%”)

Ans) In Abstract section, the following sentence was added: “The discrepancies are less than 2.0% for the mass and elastic axis locations and about 8.1% for the blade stiffness properties, as compared with the measured data.”

Before L26: Include a list (or table) of all abbreviations/acronyms and symbols used in the text and on the figures at the beginning of the document, just before the introduction. This would be helpful to the reader.

Ans) Nomenclature and abbreviations/acronyms sections were added in the revised manuscript (pages 18 – 19).

L56: after “glass fiber composite (GFRP) ”. Add “(GFRP)”

Ans) We inserted the text.

L59: “mass center (or center of gravity, CG). Add “center of gravity” into the brackets

Ans) We inserted the text.

L119: indicate the references (supplier …) of the CT scan machine

Ans) The product name was inserted in the text as: Phoenix V|tome|x L450 by Waygate Technoligies.

L172:”with a polytetrafluorethylene (PTFE, Teflon®) coating”. Indicate material type (Teflon is a commercial name)

Ans) We modified the text. Thanks for the valuable comment.

L172, 195, 246, 369, 370 and L375 (fig.12): Teflon is a trade mark. Write “Teflon®”

Ans) We inserted the trade mark symbol after Teflon.

Table 1: “Resin” and “Foam”. Indicate the types of resin (epoxy resin ?) and of foam materials

Ans) More detailed texts were included in Table 1: epoxy resin / polyurethane foam.

L209 (Table1, note a,b,c): “Elastic moduli in tension (E) and shear (G) in fiber (F) and transverse (T) directions”. Add the meaning of E, G, L, T into brackets

Ans) This was implemented in the revised version.

L213 (sub-title 2.3.2): “2D cross-sectional”.

Ans) The sub-title of the section “2.3” (not 2.3.2) was modified as “Two-dimensional (2D) …“ to avoid confusion.

Figure 7: Replace the trade mark “Kulite” by “Pressure transducer”

Ans) The wording in Figure 7 was changed.

Figure 18: Make sure that the notion EA (axial stiffness) is defined somewhere in the text (or add the definition to the note defining the abbreviations SC, TC, CG, MOI on fig 18c)

Ans) Abbreviations/acronyms section was inserted to clarify the symbols.
